# Fluid overload is a determinant for cardiac structural and functional impairments in type 2 diabetes mellitus and chronic kidney disease stage 5 not undergoing dialysis

**Byoung-Geun Han**[1], **Jun Young Lee**[1], **Mi Ryung Kim**[1], **Hanwul Shin**[1], **Jae-Seok Kim**[1], **Jae-Won Yang**[1], **Jong Yeon Kim**[2]*

1 Department of Nephrology, Yonsei University Wonju College of Medicine, Wonju, Kang-won, Korea,
2 Department of Neurosurgery, Yonsei University Wonju College of Medicine, Wonju, Kang-won, Korea

* jjongse@yonsei.ac.kr

**Data Availability Statement:** All relevant data are within the manuscript and its Supporting Information files.

## Abstract

### Background

Fluid overload is common in patients with diabetes and chronic kidney disease (DM and CKD; DMCKD) and can lead to structural and functional cardiac abnormalities including left ventricular hypertrophy (LVH) and left ventricular diastolic dysfunction (LVDD). Fluid overload represents a crucial step in the pathophysiological pathways to chronic heart failure in patients with end-stage renal disease. We evaluated the impact of fluid overload on cardiac alterations in patients with diabetes and non-dialysis-dependent CKD stage 5 (DMCKD5-ND) without intrinsic heart disease.

### Methods

Bioimpedance spectroscopy, echocardiography, and N-terminal prohormone of B-type natriuretic peptide (NT-proBNP) measurement were performed in 135 consecutive patients on the same day. Patients were divided into groups by tertiles of overhydration/extracellular water (OH/ECW) per bioimpedance spectroscopy.

### Results

Fluid balance markers including OH/ECW and NT-proBNP were significantly higher in the LVDD+LVH group. OH/ECW and its exacerbation were positively associated with the ratio between early mitral inflow and annular early diastolic velocities (E/e′ ratio) and left ventricular mass index (LVMI). The prevalence of LVH progressively increased across increasing tertiles of OH/ECW. In multiple regression analyses, OH/ECW as a continuous and categorical variable was independently associated with the E/e′ ratio and LVMI after adjustment for multiple confounding factors.

### Conclusions

Fluid overload was independently associated with LVDD and LVH in patients with DMCKD5-ND. Our study suggests that structural and functional cardiac abnormalities and

**Funding:** The author(s) received no specific funding for this work.

**Competing interests:** The authors have declared that no competing interests exist.

**Abbreviations:** cBMI, corrected body mass index; DBP, diastolic blood pressure; ECW, extracellular water; eGFR, estimated glomerular filtration rate; HbA1C, hemoglobin A1C; HDL-C, high-density lipoprotein cholesterol; hs-CRP, high-sensitivity C-reactive protein; iPTH, intact parathyroid hormone; LAD, left atrial dimension; LAVI, left atrial volume index; LDL-C, low-density lipoprotein cholesterol; LVEDD, left ventricular end-diastolic dimension; LVEF, left ventricular ejection fraction; LVEDV, left ventricular end-diastolic volume; LVDD, left ventricular diastolic dysfunction; LVH, left ventricular hypertrophy; LVMI, left ventricular mass index; NT-proBNP, N-terminal prohormone of B-type natriuretic peptide; OH, overhydration; RWT, relative wall thickness; SBP, systolic blood pressure; TBW, total body water.

volume status should be evaluated simultaneously in patients with early-stage DMCKD rather than only DMCKD5-ND, in addition to intensive blood pressure and glycemic control, regardless of evident cardiovascular disease.

## Introduction

A component of diabetic cardiomyopathy (DCM) is asymptomatic progressive structural and functional remodeling in the heart of a patient with diabetes even in the absence of preceding factors such as coronary artery disease, hypertension, and valvular heart disease. DCM includes structural changes such as left ventricular hypertrophy (LVH) and functional changes such as left ventricular diastolic dysfunction (LVDD) or left ventricular systolic dysfunction. The prevalence of LVH in the general diabetic population is known to be up to 70% [1]. LVH and LVDD may be early manifestations of unrecognized diabetic cardiac impairment [2, 3]. A number of mechanisms have been proposed to elucidate the pathophysiology of DCM [4–8].

As renal function gradually decreases in patients with diabetes, cardiac abnormalities are at a greater risk of worsening. Nonetheless, less attention has been paid to left ventricular (LV) abnormalities in patients with diabetes and chronic kidney disease. Due to the complexity of chronic kidney disease (CKD)-related cardiovascular risk factors, such as albuminuria, anemia, secondary hyperparathyroidism, and fluid overload in addition to traditional cardiovascular risk factors such as age, sex, smoking, hypertension, hyperlipidemia, and hyperglycemia, it is very difficult to clearly identify the pathophysiology of type 2 diabetes mellitus and chronic kidney disease (DMCKD). LVH and LVDD each represent a manifestation of the effects of traditional risk factors and other CKD-related cardiovascular risk factors over time and often overlap with each other in patients with end-stage renal disease (ESRD).

Patients with type 2 diabetes mellitus (T2DM) have higher risks of cardiovascular events and mortality compared with patients without T2DM [9, 10]. Cardiovascular complications are the major cause of mortality and morbidity in patients with ESRD. Chronic fluid overload in ESRD is a strong risk factor for death [11]. Notwithstanding, there are no guidelines for a comprehensive therapeutic approach for cardiac abnormalities in patients with advanced DMCKD. Most of the therapeutic goals have been derived from clinical studies involving patients with no CKD or mild CKD and they merely suggest a therapeutic approach for each of the traditional or non-traditional cardiovascular risk factors.

Assessment of fluid overload in patients with DMCKD is important not only for short-term volume management but also for the long-term prevention of cardiovascular disease. This is because fluid overload represents a crucial step in the pathophysiological pathways to chronic heart failure in ESRD patients [12, 13]. Therefore, herein, we investigated the impact of actual fluid overload on LVH and LVDD development in patients with diabetes with non-dialysis-dependent CKD stage 5 (CKD5-ND) who were free of intrinsic heart disease.

## Materials and methods

### 1. Patients and data collection

Since 2014, we have registered consecutive patients with stage 5 CKD (CKD5) to a bioimpedance cohort. All patients were hospitalized to plan their first dialysis treatment. Bioimpedance spectroscopy (BIS), echocardiography, and laboratory evaluation were performed on the same day at the time of enrollment, prior to dialysis.

Of the total cohort of patients, non-diabetic patients and those who did not undergo an echocardiographic examination were excluded from the analysis. A total of 29 patients with structural and functional cardiac abnormalities were excluded to reduce the effects of underlying heart disease that could cause LVH and/or LVDD. Patients who had a history of angina or myocardial infarction and patients who had findings of infarction on electrocardiography or had regional wall motion abnormalities on echocardiographic examination were considered as patients with coronary artery disease.

This study was conducted in accordance with the Declaration of Helsinki. This study was initiated after receiving approval (no. CR319143) from the Institutional Review Board of Yonsei University Wonju Severance Christian Hospital. All patients provided written inform consent prior to participation in the study. Therefore, the current study was a retrospective observational analysis of a prospective cohort database.

## 2. Conventional echocardiographic study

Echocardiography was performed in the harmonic imaging mode using a 3 MHz transducer and commercial ultrasound system (GE Vivid™ E9; GE Healthcare, Chicago, IL USA) prior to any dialysis treatment. The LV mass was calculated following the American Society of Echocardiography recommendations using following equation:

$$LV\ mass = 0.8 \times \{1.04 \times ([PWTd + SWTd + LVDd]^3 \times [LVEDD]^3)\} + 0.6\ g$$

where PWTd and SWTd are the posterior and septal wall thicknesses at end-diastole, respectively, and the LV end-diastolic dimension (LVEDD) is the M-mode LV dimension with the short axis view at end-diastole. To correct for body surface area, the LV mass index (LVMI) was calculated by dividing the LV mass by the body surface area (BSA), using the formula as follows: BSA = $(0.007184 \times weight^{0.425} \times height^{0.725})$ m$^2$. The relative wall thickness (RWT) was calculated by the formula: RWT = $(2 \times PWTLVEDD)$. The left atrial (LA) dimension, LA volume index, LV end-diastolic volume (LVEDV), and LV ejection fraction (LVEF) were measured using the biplane modified Simpson's rule, according to the previously mentioned recommendations. Transmitral inflow velocities were measured using pulsed-wave Doppler in the apical four-chamber view with the sample volume placed at the mitral valve leaflet tips. Transmitral early diastolic (E wave) velocities were measured. Tissue Doppler imaging in the apical four-chamber view was used to measure LV myocardial velocities, with the sample volume placed at the septal mitral annulus. We measured the peak early (e′) diastolic mitral annular velocity and calculated the E/e′ ratio [14]. LVH was defined as an LVMI >95 g/m$^2$ in females and >115 g/m$^2$ in males [15]. LVDD was defined as an E/e′ ratio >15 [16]. Echocardiography was performed by trained cardiologists who were completely blinded to the patient information.

## 3. Assessment of the volume status

Whole-body BIS was performed using the BCM system (Body Composition Monitoring™, Fresenius Medical Care AG & Co., Bad Homburg vor der Höhe, Germany) prior to any dialysis treatment. Patients were in supine positions. Disposable electrode patches placed on the wrist and ankle were used for all measurements. Measurements were performed in the absence of metal and electronic devices on patients to minimize disruption. BCM utilizes alternating electric currents across 50 discrete frequencies covering the frequency spectrum from 5 to 1,000 kHz and measures each current's impedance. A three compartment BIS model separates the body weight into normally hydrated lean tissue mass, normally hydrated adipose tissue mass, and fluid overload which is commonly described as the overhydration (OH)

compartment [17]. Extracellular water (ECW), intracellular water, and total body water (TBW) were automatically calculated. The OH, which can be calculated from the difference between the actual measured ECW and the normally expected ECW, can be positive or negative [18]. Relative overhydration can be represented as OH/ECW. OH level, OH/ECW, ECW/TBW, and N-terminal prohormone of B-type natriuretic peptide (NT-proBNP) level were used as markers of fluid balance. In this study, volume status was treated as OH/ECW in the main analysis. The patient's body mass index was recalculated for a corrected body mass index (cBMI) by considering fluid overload using the formula as follows: cBMI ($kg/m^2$) = (body weight–OH)/height$^2$.

## 4. Laboratory evaluations

All laboratory studies were performed before the first dialysis application. The high-sensitivity C-reactive protein (hs-CRP) level was measured using a Cobas 8000 Modular Analyzer (Roche Diagnostics GmbH, Mannheim, Germany). The normal range for the hs-CRP is below 0.3 g/dL (3 g/L). The estimated glomerular filtration rate (eGFR) was calculated using the formula developed for the Modification of Diet in Renal Disease (MDRD) Study that is based on serum creatinine. The NT-proBNP level was measured using an electrochemiluminescence immuno-assay (ECLIA) using the Modular Analytics E170 System (Roche Diagnostics, GmbH, Mannheim, Germany). The analytical measurement range for NT-proBNP was from 5 to 35,000 pg/mL. The patients were divided into three groups in accordance with the tertiles of NT-proBNP for the regression analysis.

## 5. Statistical analysis

Categorical variables were reported as frequencies and percentages, and continuous variables were reported as means with standard deviations or medians with interquartile ranges, as appropriate. All patients were classified into four groups according to the phenotypes of cardiac impairment (control patients with CKD5 but without cardiac impairment, and those with LVDD, LVH, and LVDD+LVH). All patients were also divided into three groups by tertile of OH/ECW for an analysis. Patient characteristics between groups were tested with a chi-squared test, Fisher's exact test, analysis of variance (ANOVA) with post hoc Bonferroni correction, and a Kruskal–Wallis test. A linear-by-linear association method and a Jonckheere-Terpstra trend test were used for analyzing trends in the OH/ECW tertiles. We first performed stepwise linear regression analyses to identify the potential determinants of E/e′ ratio and LVMI adjusted for age, hs-CRP, albumin, phosphorus, and eGFR. We then performed multiple linear regression analyses to explore the association of E/e′ ratio and LVMI with the identified determinants. OH/ECW was considered as both a continuous and a categorical variable in the multivariate linear regression analyses. All analyses were performed using IBM SPSS Statistics software (version 23.0; IBM Corporation, Armonk, NY USA). Graphs were generated with Prism software (version 5.02; GraphPad Software, San Diego, CA USA). Statistically significant differences were defined as those having $P$-values <0.05.

## Results

### 1. Characteristics of the study patients

After the application of the exclusion criteria, a total of 135 patients (control patients with CKD5, n = 35; LVDD, n = 21; LVH, n = 30; LVDD+LVH, n = 49) were analyzed (Fig 1). The mean age of male and female patients was 59.53 ± 12.29 years and 61.11 ± 10.80 years, respectively. Females accounted for 40% (n = 54) of all patients. The mean eGFR was 7.11 ± 2.39 mL/

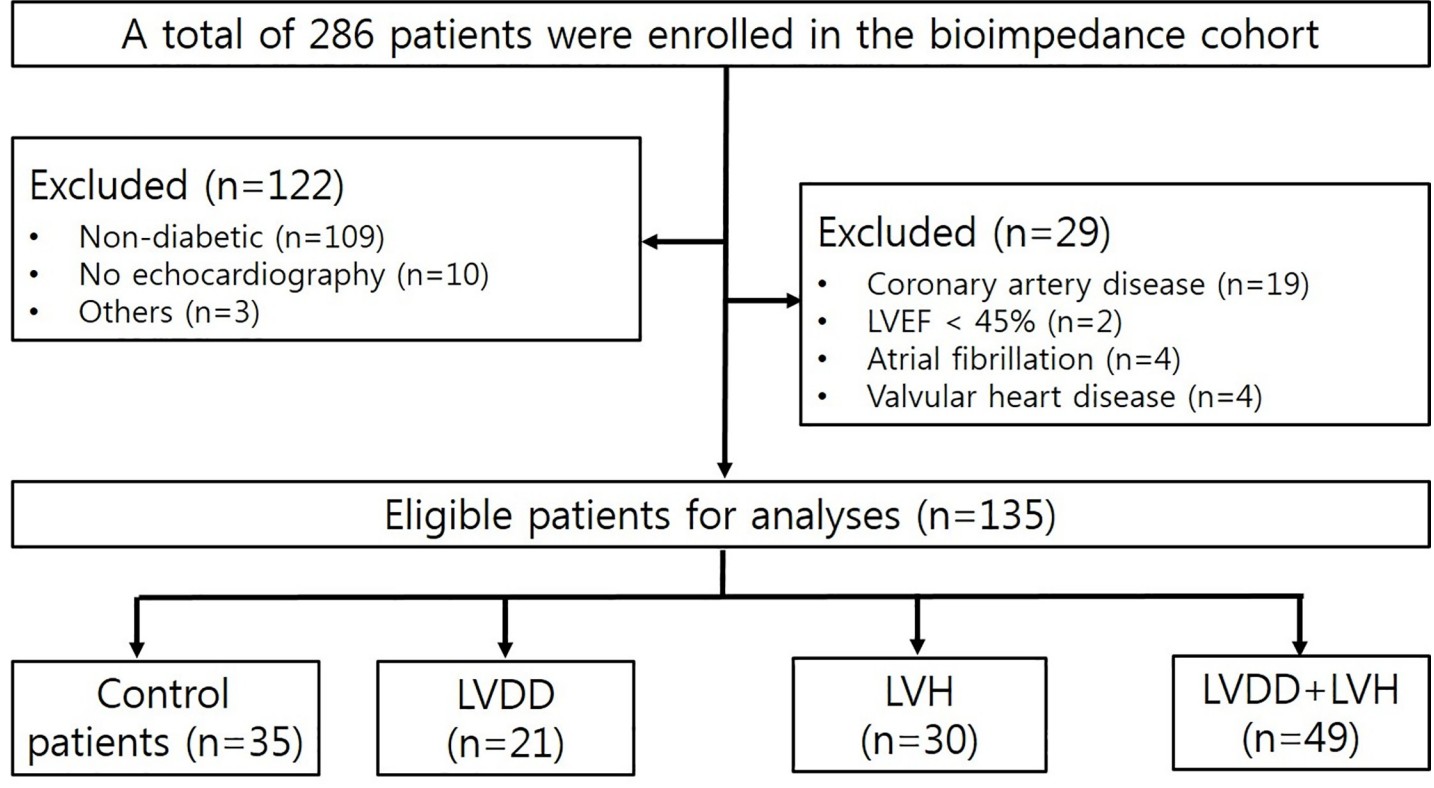

**Fig 1. Flow diagram of patient selection in this study.**

min/1.73 m². In the present study, fluid overload was present in 101 patients (74.81%) when defined as OH/ECW over 7%, whereas severe fluid overload defined as OH/ECW over 15% was present in 77 patients (57.04%).

The clinical characteristics of each group according to the left ventricular abnormalities are presented in Table 1. Patients with diastolic dysfunction and hypertrophy had lower levels of eGFR and albumin, while markers of fluid balance such as OH, OH/ECW, ECW/TBW, and NT-proBNP were significantly higher than in any of the other groups. The median (interquartile range [IQR]) for OH/ECW and NT-proBNP were 17.40% (6.90–28.10%) and 2,842 (674–8,029) pg/mL, respectively.

The clinical characteristics of the patients according to the tertiles of OH/ECW are presented in Table 2. Compared with patients in the first tertile of OH/ECW, the mean age for those in the third tertile was significantly lower. The eGFR, hemoglobin, albumin, and calcium levels were lower for those in the third tertile than for those in the first tertile. There was a trend of higher levels of phosphorus, hs-CRP, and NT-proBNP across increasing tertiles of OH/ECW. Exacerbation of the OH/ECW was significantly associated with echocardiographic findings including LAD, LAVI, E/e′ ratio, and LVMI. By running a post hoc test, the E/e′ ratio was significantly higher for those in the third tertile than in the first and second tertiles. Second and third tertile groups had significantly higher levels of LVMI than those in the first tertile group. There was no difference in LVEF between the three groups (Fig 2).

OH/ECW was positively associated with LA dimension (r = 0.236; P = 0.007), LAVI (r = 0.408; P < 0.001), E/e′ ratio (r = 0.344; P < 0.001), LVEDV (r = 0.178; P = 0.042), RWT (r = 0.172; P = 0.049), and LVMI (r = 0.359; P < 0.001). While OH/ECW was not significantly

**Table 1. Comparisons of demographics, serum chemistry, echocardiographic findings, and volume status between the groups according to the left ventricular abnormalities.**

| Variables | Control patients with CKD5 (n = 35) | LVDD (n = 21) | LVH (n = 30) | LVDD+LVH (n = 49) | P-value |
|---|---|---|---|---|---|
| Age, years | 59.89±11.29 | 60.10±11.55 | 59.30±11.17 | 60.92±12.64 | 0.944 |
| <65 years | 21 (25.0%) | 14 (16.7%) | 20 (23.8%) | 29 (34.5%) | 0.874 |
| ≥65 years | 14 (27.5%) | 7 (13.7%) | 10 (19.6%) | 20 (39.2%) | |
| Sex | | | | | |
| Male | 27 (33.3%) | 14 (17.3%) | 20 (24.7%) | 20 (24.7%) | 0.005 |
| Female | 8 (14.8%) | 7 (13.0%) | 10 (18.5%) | 29 (53.7%) | |
| SBP, mmHg | 140.19±22.73 | 145.52±16.84 | 142.21±15.74 | 149.83±15.74 | 0.098 |
| DBP, mmHg | 83.32±11.54 | 79.14±10.33 | 77.90±8.61 | 79.96±9.78 | 0.178 |
| cBMI, kg/m$^2$ | 23.44±4.11 | 25.24±4.09 | 24.06±4.38 | 24.08±4.20 | 0.497 |
| LAD, cm | 4.33±0.42 | 4.68±0.49 | 4.61±0.40 | 4.84±0.44 | <0.001 |
| LAVI, mL/m$^2$ | 30.66±6.46 | 34.19±8.23 | 37.60±7.37 | 43.14±10.97 | <0.001 |
| E/e′ ratio | 11.60±2.53 | 17.35±1.31 | 13.01±1.96 | 21.00±5.27 | <0.001 |
| LVEDD, cm | 5.09±0.56 | 5.34±0.43 | 5.53±0.53 | 5.47±0.48 | 0.002 |
| LVEDV, mL | 127.69±27.96 | 138.91±25.36 | 152.10±33.45 | 145.18±36.38 | 0.018 |
| LVMI, g/m$^2$ | 90.97±14.43 | 99.76±10.22 | 128.17±16.83 | 127.41±19.42 | <0.001 |
| RWT | 0.35±0.06 | 0.33±0.04 | 0.37±0.05 | 0.37±0.06 | 0.037 |
| LVEF, % | 63.13±4.73 | 62.67±5.56 | 63.87±6.26 | 63.39±5.01 | 0.878 |
| NT-proBNP, pg/mL* | 712 (293–6,334) | 886 (565–2,847) | 3,299 (1,186–7,305) | 7,413 (2,589–19,247) | <0.001 |
| hs-CRP, mg/dL | 1.62±2.88 | 0.54±1.05 | 0.95±2.74 | 1.76±3.74 | 0.395 |
| iPTH, pg/mL | 228.17±114.92 | 315.63±198.96 | 285.31±137.79 | 282.15±168.87 | 0.190 |
| HbA1C, % | 7.22±2.08 | 7.21±2.14 | 6.63±1.10 | 6.91±1.41 | 0.514 |
| Hemoglobin, g/dL | 9.45±1.35 | 9.30±1.26 | 9.03±1.33 | 8.85±1.13 | 0.165 |
| Total protein, g/dL | 6.30±0.86 | 6.54±0.86 | 5.87±0.70 | 5.88±0.71 | 0.002 |
| Albumin, g/dL | 3.54±0.58 | 3.78±0.51 | 3.29±0.55 | 3.20±0.51 | <0.001 |
| Total cholesterol, mg/dL | 143.57±41.61 | 140.19±36.00 | 148.35±41.03 | 152.10±44.38 | 0.671 |
| HDL-C, mg/dL | 39.24±14.64 | 36.57±13.04 | 39.14±11.56 | 38.55±12.54 | 0.887 |
| LDL-C, mg/dL | 76.00±35.44 | 76.57±30.59 | 85.52±36.45 | 87.32±38.83 | 0.446 |
| Triglyceride, mg/dL | 143.00±71.47 | 138.43±56.35 | 127.45±48.31 | 129.33±45.65 | 0.617 |
| Calcium, mg/dL | 8.04±0.88 | 8.21±0.98 | 7.59±1.02 | 7.57±0.85 | 0.012 |
| Phosphate, mg/dL | 5.54±1.65 | 5.40±1.03 | 6.11±1.60 | 6.22±1.28 | 0.053 |
| eGFR, mL/min/1.73 m$^2$ | 8.01±2.63 | 7.55±2.37 | 7.02±2.35 | 6.34±2.01 | 0.011 |
| OH, liters | 2.52±3.11 | 2.07±1.94 | 3.53±2.40 | 4.98±3.72 | <0.001 |
| OH/ECW, % | 12.88±14.37 | 11.71±10.70 | 18.57±9.60 | 25.06±14.25 | <0.001 |
| ECW/TBW | 0.49±0.04 | 0.50±0.04 | 0.51±0.03 | 0.54±0.04 | <0.001 |

* Kruskal–Wallis test; median (interquartile range).

cBMI, corrected body mass index; DBP, diastolic blood pressure; ECW, extracellular water; eGFR, estimated glomerular filtration rate; HbA1C, hemoglobin A1C; HDL-C, high-density lipoprotein cholesterol; hs-CRP, high-sensitivity C-reactive protein; iPTH, intact parathyroid hormone; LAD, left atrial dimension; LAVI, left atrial volume index; LDL-C, low-density lipoprotein cholesterol; LVEDD, left ventricular end-diastolic dimension; LVEF, left ventricular ejection fraction; LVEDV, left ventricular end-diastolic volume; LVDD, left ventricular diastolic dysfunction; LVH, left ventricular hypertrophy; LVMI, left ventricular mass index; NT-proBNP, N-terminal prohormone of B-type natriuretic peptide; OH, overhydration; RWT, relative wall thickness; SBP, systolic blood pressure; TBW, total body water.

associated with LVEDD, LVEDV and LVEF, OH/ECW was positively associated with hs-CRP (r = 0.219; *P* = 0.017) and inversely associated with cBMI (r = -0.280; *P* = 0.001), eGFR (r = -0.302; *P* < 0.001), and serum calcium levels (r = -0.367; *P* < 0.001). There was a marginally significant association between OH/ECW and SBP (r = 0.171; *P* = 0.051).

**Table 2. Comparisons of demographics, serum chemistry, echocardiographic findings, and volume status according to the OH/ECW tertiles.**

| Variables | OH/ECW (%) | | | P-value | P for trend# |
|---|---|---|---|---|---|
| | Tertile 1 | Tertile 2 | Tertile 3 | | |
| Age, years | 63.12±9.07 | 60.46±11.30 | 56.11±13.54 | 0.018 | 0.005 |
| <65 years | 24 (28.9%) | 25 (30.1%) | 34 (41.0%) | 0.063 | 0.038 |
| ≥65 years | 19 (39.6%) | 19 (39.6%) | 10 (20.8%) | | |
| Sex | | | | | |
| Male | 23 (29.1%) | 32 (40.5%) | 24 (30.4%) | 0.118 | 0.931 |
| Female | 20 (38.5%) | 12 (23.1%) | 20 (38.5%) | | |
| SBP, mmHg | 139.14±18.60 | 145.98±18.98 | 149.46±17.23 | 0.031 | 0.012 |
| DBP, mmHg | 77.11±8.99 | 81.07±10.00 | 82.48±10.95 | 0.039 | 0.027 |
| cBMI, kg/m$^2$ | 25.03±3.62 | 24.87±4.41 | 22.37±4.05 | 0.003 | 0.001 |
| LAD, cm | 4.46±0.43 | 4.64±0.45 | 4.75±0.49 | 0.017 | 0.001 |
| LAVI, mL/m$^2$ | 32.05±8.32 | 36.89±8.12 | 42.11±10.56 | <0.001 | <0.001 |
| E/e′ ratio | 14.72±4.09 | 15.18±4.72 | 18.81±6.36 | <0.001 | 0.002 |
| LVEDD, cm | 5.22±0.44 | 5.50±0.62 | 5.40±0.49 | 0.048 | 0.124 |
| LVEDV, mL | 129.76±32.00 | 151.36±33.58 | 142.30±30.67 | 0.008 | 0.107 |
| LVMI, g/m$^2$ | 102.47±15.67 | 119.23±24.81 | 119.82±24.54 | <0.001 | <0.001 |
| RWT | 0.34±0.05 | 0.36±0.06 | 0.37±0.06 | 0.164 | 0.029 |
| LVEF, % | 63.26±4.45 | 62.55±5.03 | 64.11±5.90 | 0.364 | 0.560 |
| NT-proBNP, pg/mL* | 632 (380–1,555) | 3,137 (1,056–7,350) | 8,025 (4,398–18,736) | <0.001 | <0.001 |
| hs-CRP, mg/dL | 0.56±1.01 | 1.33±3.10 | 2.01±3.86 | 0.107 | 0.004 |
| iPTH, pg/mL | 290.59±168.81 | 286.37±170.41 | 253.71±131.23 | 0.496 | 0.333 |
| HbA1C, % | 7.40±2.00 | 6.87±1.38 | 6.72±1.61 | 0.174 | 0.055 |
| Hemoglobin, g/dL | 9.61±1.35 | 9.06±1.19 | 8.79±1.02 | 0.006 | 0.005 |
| Total protein, g/dL | 6.74±0.68 | 6.00±0.55 | 5.54±0.72 | <0.001 | <0.001 |
| Albumin, g/dL | 3.82±0.43 | 3.35±0.42 | 3.02±0.55 | <0.001 | <0.001 |
| Total cholesterol, mg/dL | 147.21±39.91 | 138.96±33.46 | 154.55±49.80 | 0.217 | 0.627 |
| HDL-C, mg/dL | 39.00±13.38 | 36.12±11.13 | 40.54±14.20 | 0.280 | 0.592 |
| LDL-C, mg/dL | 79.95±33.27 | 77.21±31.15 | 89.28±43.45 | 0.276 | 0.477 |
| Triglyceride, mg/dL | 144.12±61.93 | 131.96±53.75 | 123.07±49.05 | 0.206 | 0.120 |
| Calcium, mg/dL | 8.24±1.00 | 7.63±0.89 | 7.49±0.78 | <0.001 | <0.001 |
| Phosphate, mg/dL | 5.52±1.43 | 6.02±1.39 | 6.10±1.46 | 0.126 | 0.023 |
| eGFR, mL/min/1.73 m$^2$ | 8.12±2.29 | 6.82±2.39 | 6.42±2.03 | 0.002 | 0.001 |
| OH, liters | 0.52±0.76 | 2.96±0.76 | 7.12±2.81 | <0.001 | <0.001 |
| OH/ECW, % | 3.24±5.03 | 17.01±3.29 | 34.37±7.44 | <0.001 | <0.001 |
| ECW/TBW | 0.47±0.03 | 0.51±0.02 | 0.56±0.03 | <0.001 | <0.001 |

* Kruskal–Wallis test; median (interquartile range)

# P-values obtained by the linear-by-linear association method or Jonckheere–Terpstra test.

OH/ECW tertiles 1, 2, and 3 correspond to <10.60, 10.60–24.68, and >24.68%, respectively.

cBMI, corrected body mass index; DBP, diastolic blood pressure; ECW, extracellular water; eGFR, estimated glomerular filtration rate; HbA1C, hemoglobin A1C; HDL-C, high-density lipoprotein cholesterol; hs-CRP high-sensitivity C reactive protein; iPTH, intact parathyroid hormone; LAD, left atrial dimension; LAVI, left atrial volume index; LDL-C, low-density lipoprotein cholesterol; LVEDD, left ventricular end-diastolic dimension; LVEF, left ventricular ejection fraction; LVEDV, left ventricular end-diastolic volume; LVH, left ventricular hypertrophy; LVMI, left ventricular mass index; NT-proBNP, N-terminal prohormone of B-type natriuretic peptide; OH, overhydration; RWT, relative wall thickness; SBP, systolic blood pressure; TBW, total body water.

LVDD was present in 70 patients (51.85%) (S1 Table). LVH was present in 79 patients (58.52%) (S2 Table). The prevalence of LVDD was 48.8%, 38.6%, and 68.2% and for LVH was 35.0%, 65.9%, and 72.7% in OH/ECW tertiles 1, 2, and 3, respectively. The prevalence of LVH

progressively increased across increasing tertiles of OH/ECW (*P* for trend <0.001) (Fig 3). However, the *P*-value for trend of LVDD prevalence was not significant (*P* for trend = 0.070).

## 2. Determining factors of E/e′ ratio and LVMI

Stepwise linear regression analyses showed that cBMI, OH/ECW, being female, and SBP were shown to be significantly associated with E/e′ ratio, after adjustment for clinical confounding

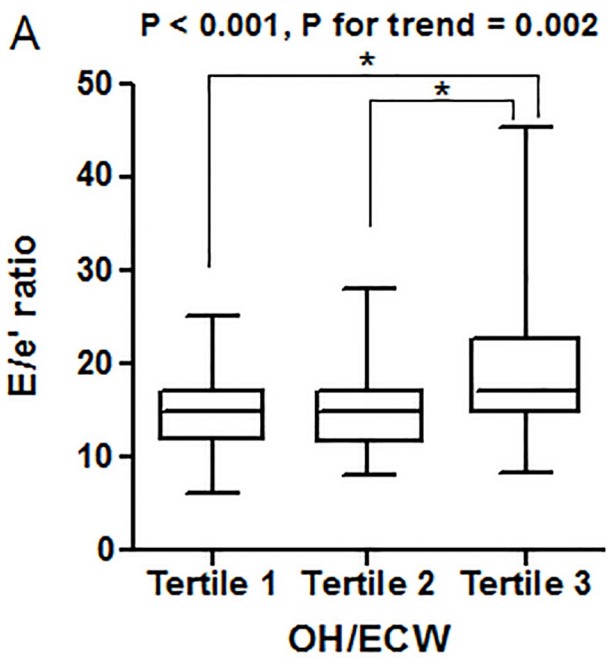
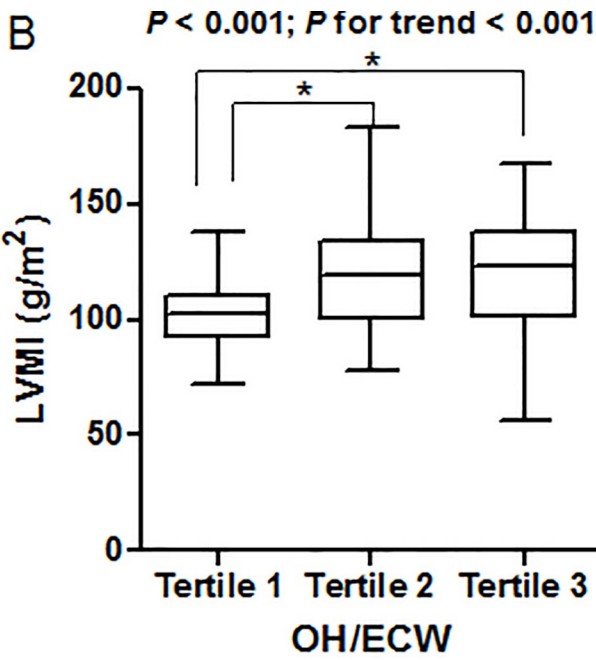
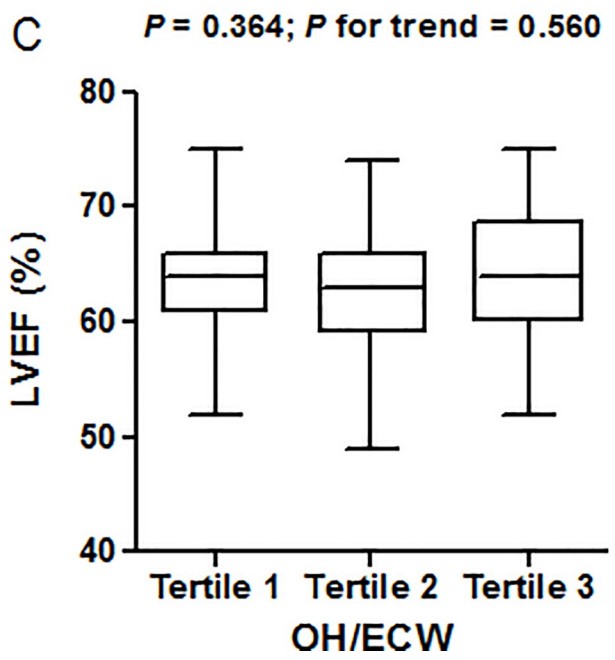
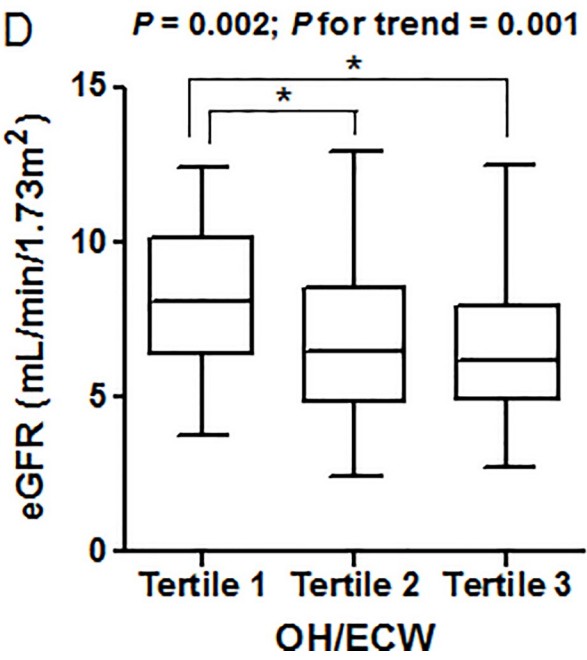

**Fig 2. Left ventricular structural and functional alterations and glomerular filtration rate according to the tertile distribution of OH/ECW.**

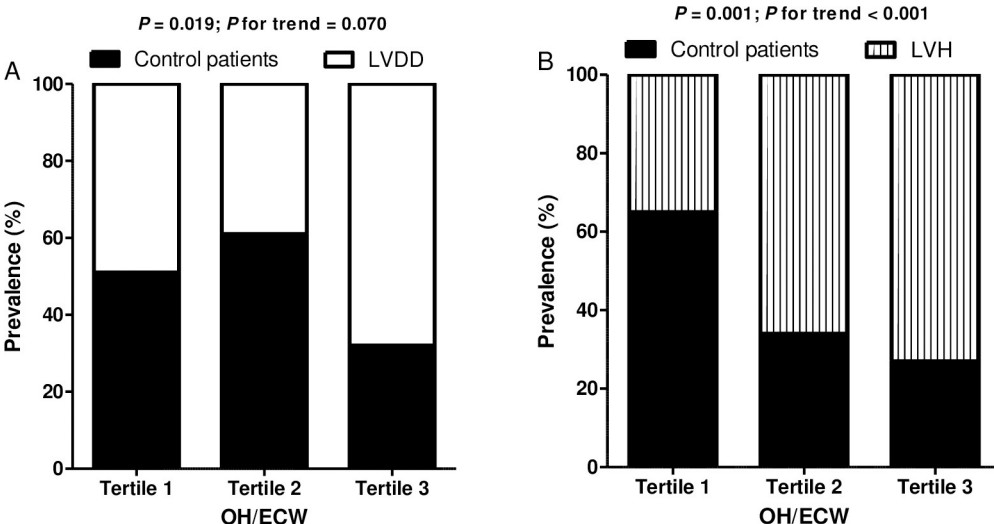

**Fig 3. Comparison of the percentage distribution of subjects according to the tertile distribution of OH/ECW.** A) LVDD; chi-squared test, $P = 0.019$, B) LVH; chi-squared test, $P = 0.001$.

factors including age, hs-CRP, albumin, phosphorus, and eGFR. Meanwhile, cBMI, OH/ECW, and serum calcium levels were significantly associated with LVMI after adjustment for confounding factors (S3 Table).

## 3. Multiple linear regression analyses

When evaluating OH/ECW as a continuous variable in a multiple linear regression analysis, the association of OH/ECW with E/e′ ratio remained statistically significant in all three models. When evaluating OH/ECW and NT-proBNP as categorical variables for further analysis, both showed significant β coefficients in the third tertile compared to the reference group (Table 3). In addition, similar associations between OH/ECW as a continuous variable and LVMI remained statistically significant in fully adjusted models. When evaluating OH/ECW and NT-proBNP as categorical variables, both parameters also showed significant associations with LVMI (Table 4). All of these associations suggest that OH/ECW and NT-proBNP are contributing factors for LVDD and LVH development in patients with diabetes and CKD5-ND who are free of intrinsic heart disease.

## Discussion

DCM was introduced by Rubler et al. and is known as a risk factor for heart failure [19]. Prior to the emergence of clinical heart failure, DCM is associated with LVH, cardiac re-modeling, advancing diastolic dysfunction, and subclinical left ventricular systolic impairment [20].

In-depth studies have been conducted on who should undergo screening in relation to the progression of DCM in patients with diabetes with normal kidney function. However, there has not been much research on the diagnostic approach for patients with DMCKD only. Cardiovascular disease is the leading cause of complications in diabetes. CKD is another major risk factor for cardiovascular morbidity and mortality regardless of the cause. Given that diabetes is the most common cause of ESRD, cardiovascular complications are the major cause of mortality and morbidity in patients with DMCKD. As the pathophysiology of DMCKD is so

**Table 3. Factors independently associated with E/e′ ratio.**

| | Unadjusted | Model 1 | Model 2 | Model 3 |
| --- | --- | --- | --- | --- |
| | B (95% CI) | B (95% CI) | B (95% CI) | B (95% CI) |
| **OH/ECW analyzed as a continuous variable** | | | | |
| OH/ECW, % | 0.135 (0.071, 0.199) | 0.148 (0.084, 0.211) | 0.152 (0.088, 0.216) | 0.134 (0.057, 0.211) |
| **OH/ECW analyzed in tertiles** | | | | |
| OH/ECW, % | | | | |
| tertile 1 | Reference | Reference | Reference | Reference |
| tertile 2 | 0.458 (-1.729, 2.644) | 1.168 (-0.960, 3.297) | 0.783 (-1.262, 2.828) | 0.084 (-2.238, 2.405) |
| tertile 3 | 4.083 (1.896, 6.269) | 4.454 (2.295, 6.613) | 4.404 (2.206, 6.601) | 3.539 (0.973, 6.105) |
| **NT-proBNP analyzed in tertiles** | | | | |
| NT-proBNP, pg/mL | | | | |
| tertile 1 | Reference | Reference | Reference | Reference |
| tertile 2 | 1.820 (-0.352, 3.991) | 1.466 (-0.642, 3.574) | 1.170 (-0.911, 3.250) | 0.819 (-1.595, 3.234) |
| tertile 3 | 3.781 (1.610, 5.953) | 3.756 (1.662, 5.850) | 3.663 (1.537, 5.789) | 2.938 (0.199, 5.676) |

Model 1: Adjusted for age and sex.

Model 2: Adjusted for age, sex, cBMI, and SBP.

Model 3: Adjusted for age, sex, cBMI, SBP, calcium, eGFR, and hs-CRP.

OH/ECW tertiles 1, 2, and 3 correspond to <10.60, 10.60–24.68, and >24.68%, respectively.

NT-proBNP tertiles 1, 2, and 3 correspond to <1,143, 1,143–7,087, and >7,087 pg/mL, respectively.

B, β coefficient; cBMI, corrected body mass index; CI, confidence interval; ECW, extracellular water; eGFR, estimated glomerular filtration rate; hs-CRP, high-sensitivity C reactive protein; NT-proBNP, N-terminal prohormone B-type natriuretic peptide; OH, overhydration; SBP, systolic blood pressure.

**Table 4. Factors independently associated with left ventricular mass index.**

| | Unadjusted | Model 1 | Model 2 | Model 3 |
| --- | --- | --- | --- | --- |
| | B (95% CI) | B (95% CI) | B (95% CI) | B (95% CI) |
| **OH/ECW analyzed as a continuous variable** | | | | |
| OH/ECW, % | 0.604 (0.330, 0.878) | 0.684 (0.402, 0.966) | 0.725 (0.430, 1.020) | 0.525 (0.157, 0.893) |
| **OH/ECW analyzed in tertiles** | | | | |
| OH/ECW, % | | | | |
| tertile 1 | Reference | Reference | Reference | Reference |
| tertile 2 | 16.762 (7.374, 26.151) | 16.389 (6.882, 25.986) | 15.610 (6.060, 25.159) | 12.194 (1.374, 23.013) |
| tertile 3 | 17.353 (7.965, 26.742) | 14.412 (9.218, 28.505) | 19.113 (8.852, 29.373) | 13.225 (1.266, 25.183) |
| **NT-proBNP analyzed in tertiles** | | | | |
| NT-proBNP, pg/mL | | | | |
| tertile 1 | Reference | Reference | Reference | Reference |
| tertile 2 | 17.911 (8.919, 26.902) | 18.591 (9.639, 27.544) | 19.264 (10.213, 28.315) | 16.494 (5.881, 27.107) |
| tertile 3 | 21.155 (12.164, 30.146) | 21.353 (12.459, 30.248) | 23.098 (13.849, 32.347) | 15.981 (3.944, 28.018) |

Model 1: Adjusted for age and sex.

Model 2: Adjusted for age, sex, cBMI, and SBP.

Model 3: Adjusted for age, sex, cBMI, SBP, calcium, eGFR, and hs-CRP.

OH/ECW tertiles 1, 2, and 3 correspond to <10.60, 10.60–24.68, and >24.68%, respectively.

NT-proBNP tertiles 1, 2, and 3 correspond to <1,143, 1,143–7,087, and >7,087 pg/mL, respectively.

B, β coefficient; cBMI, corrected body mass index; CI, confidence interval; ECW, extracellular water; eGFR, estimated glomerular filtration rate; hs-CRP, high-sensitivity C reactive protein; NT-proBNP, N-terminal prohormone B-type natriuretic peptide; OH, overhydration; SBP, systolic blood pressure.

complex and most cases have more than one risk factor [21], it may be difficult to determine the exact cause of cardiac impairment in patients with advanced DMCKD.

LVDD is regarded as the first functional change of subclinical cardiac alterations in patients with diabetes without renal insufficiency [3, 22–24], and the prevalence of LVH in diabetes is higher [1]. LVDD is also common in CKD regardless of the cause [25, 26]. In patients with CKD, the mechanism of LVDD is complex and mainly associated with LVH, which is a physiological adaptive response to pressure and/or volume overload [27, 28]. LVH is intrinsically arrhythmogenic, leads to diastolic heart failure, and causes ischemia and sudden death [29]. Although hypertension is not a reliable biomarker of volume overload in patients with ESRD, the effect of volume overload on blood pressure is crucial. A higher level of ECW was an independent determinant of both resistant and uncontrolled hypertension during CKD [30]. Blood pressure still plays a major role in inducing LV remodeling in CKD. Therefore, our results support that fluid overload is a considerable risk factor for functional and structural LV alteration. The poor prognosis of fluid overload is mainly explained by the link with cardiovascular effects such as LVH, LV systolic and diastolic dysfunction, pulmonary hypertension, and increased aortic stiffness [31]. Therefore, the need for clinical evaluation to diagnose cardiac alterations in the early stages of CKD is emphasized [32].

In our study, there was a trend of higher levels of E/e′ ratio, LVMI, and NT-proBNP across increasing tertiles of OH/ECW. Relative overhydration (OH/ECW) was independently associated with E/e′ ratio and LVMI in a multiple linear regression. NT-proBNP, a marker for fluid overload in our study, showed significantly different levels between patients with and without cardiac abnormalities, such as LVDD and LVH, respectively. It is unclear whether NT-proBNP itself reflects volume status or whether it is a byproduct of structural and functional alterations of the myocardium due to fluid overload in our study, even though NT-proBNP is known to be associated with LVDD [33] and LVH [34]. The diagnostic value of NT-proBNP in patients with CKD is limited because decreased renal function affects NT-proBNP levels and there are no optimal cut-off values for diagnosis confirmation [35]. In addition, NT-proBNP levels are influenced by the presence of diabetes [36].

In recent years, many studies on the cardioprotective effects of a novel class of antidiabetic agents have been published. A representative drug of interest is sodium–glucose cotransporter-2 (SGLT-2) inhibitor. Proposed pathophysiologic mechanisms of SGLT-2 inhibitors for cardioprotective effects are natriuretic effect, weight loss, reduced arterial stiffness, and improvement in ventricular loading conditions [37, 38]. Reducing cardiac preload and afterload results in lowering left ventricular structural and functional changes [39]. Prior to SGLT-2 inhibitor use, no other diuretic demonstrated clear cardiorenal protective effects in patients with diabetes although they reduced blood pressure or volume overload. None of the patients in our study were given any of these drugs because the safety of these drugs for patients with late stage of DMCKD has not yet been established.

This study includes several limitations. We could not confirm that LVDD and LVH were independently related to the duration of diabetes and/or diabetic kidney disease in the multiple linear regression model. Long-term effects of volume overload and uncontrolled blood pressure instability on the induction of cardiac impairments were not evaluated. As our patients were accompanied by complicated combinations of diabetes, hypertension, and kidney disease as well as medications, a clinically important point would be the longitudinal change of cardiac alterations in proportion to the progression of the disease. Despite these limitations, the strengths of our study include that all patients were free of intrinsic heart disease. In addition, analyses were adjusted for multiple confounding factors which were highly associated with OH/ECW, including cBMI, eGFR, hs-CRP, and serum calcium levels.

## Conclusions

Our study suggests that strict volume controls are the cornerstone of effective treatment and prevention of the aforementioned cardiac impairments in patients with DMCKD5. Moreover, our study suggests that evaluation of the structural and functional cardiac abnormalities and volume status should be performed simultaneously in patients with early-stage DMCKD rather than only in patients with DMCKD5, in addition to intensive blood pressure and glycemic control, regardless of evident cardiovascular disease.

## Supporting information

**S1 Table. Comparison of demographics, serum chemistry, echocardiographic findings, and volume status between patients with and without left ventricular diastolic dysfunction.**
(DOCX)

**S2 Table. Comparison of demographics, serum chemistry, echocardiographic findings, and volume status between patients with and without left ventricular hypertrophy.**
(DOCX)

**S3 Table. Stepwise multiple linear regression of variables associated with E/e′ ratio and left ventricular mass index.**
(DOCX)

## Author Contributions

**Conceptualization:** Byoung-Geun Han, Jong Yeon Kim.

**Data curation:** Byoung-Geun Han, Jun Young Lee.

**Formal analysis:** Byoung-Geun Han, Jae-Won Yang.

**Funding acquisition:** Byoung-Geun Han, Jae-Won Yang.

**Investigation:** Byoung-Geun Han, Jae-Won Yang.

**Methodology:** Byoung-Geun Han, Jun Young Lee, Hanwul Shin.

**Project administration:** Byoung-Geun Han, Jae-Won Yang.

**Resources:** Byoung-Geun Han, Jae-Seok Kim, Jae-Won Yang.

**Software:** Byoung-Geun Han, Jun Young Lee.

**Supervision:** Jae-Won Yang.

**Validation:** Byoung-Geun Han, Jae-Won Yang.

**Visualization:** Byoung-Geun Han, Jun Young Lee.

**Writing – original draft:** Byoung-Geun Han, Mi Ryung Kim.

**Writing – review & editing:** Byoung-Geun Han, Jae-Won Yang, Jong Yeon Kim.

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
