## [Decision Letter · Decision Letter 0]

14 May 2020

PONE-D-20-08628

Fluid overload is a determinant for cardiac structural and functional impairments in type 2 diabetes mellitus and chronic kidney disease stage 5 not undergoing dialysis

PLOS ONE

Dear Prof. Han,

Thank you for submitting your manuscript to PLOS ONE. After careful consideration, we feel that it has merit but does not fully meet PLOS ONE’s publication criteria as it currently stands. Therefore, we invite you to submit a revised version of the manuscript that addresses the points raised during the review process.

ACADEMIC EDITOR: All issues raised by reviewers are required.

We would appreciate receiving your revised manuscript by Jun 28 2020 11:59PM. To enhance the reproducibility of your results, we recommend that if applicable you deposit your laboratory protocols in protocols.io, where a protocol can be assigned its own identifier (DOI) such that it can be cited independently in the future. For instructions see: http://journals.plos.org/plosone/s/submission-guidelines#loc-laboratory-protocols

We look forward to receiving your revised manuscript.

Kind regards,

Vincenzo Lionetti, M.D., PhD

Academic Editor

PLOS ONE

Journal Requirements:

a) Did participants provide their written or verbal informed consent to participate in this study?

Reviewers' comments:

Reviewer's Responses to Questions

**Comments to the Author**

1. Is the manuscript technically sound, and do the data support the conclusions?

Reviewer #1: Yes

Reviewer #2: Yes

2. Has the statistical analysis been performed appropriately and rigorously? 

Reviewer #1: Yes

Reviewer #2: I Don't Know

3. Have the authors made all data underlying the findings in their manuscript fully available?

Reviewer #1: Yes

Reviewer #2: Yes

4. Is the manuscript presented in an intelligible fashion and written in standard English?

Reviewer #1: Yes

Reviewer #2: Yes

5. Review Comments to the Author

Reviewer #1: Dear Authors,

it is an interesting manuscript that offers sound information about the effects of fluids overload on the heart in type2 DM.

it is well written and straightforward.

I have only a couple of observations:

In abstract you write that measurements were performed "on the same day". In methods it is not clear when the measurement were done (prior dyalisis). May you better explain it ? What the story before the day of measurement ? how long patients have suffered DM, how long the story of CKD ?

Another aspect needs some information. Volume status was calculated with bioimpedenzometry that is affected by BMI. You somehow corrected this, but it could be interesting to have the BMI of the evaluated patients and a discussion on the possible bias caused by BMI in the analysis. There were differences between patients with low and high BMI ?

minor: in § "assessment of the volume status" explanation of BIS and BCM are lacking.

Reviewer #2: In this paper, the authors want to investigate the association between fluid overload and left ventricular hypertrophy and left ventricular diastolic dysfunction in patient with DM type 2 and stage 5 chronic kidney disease. The manuscript is fairly well written, flow and readability are good. The rationale of the paper is clear and the analysis is sound. I consider the manuscript a good piece of research that improves the understanding of an interesting subject.

6. PLOS authors have the option to publish the peer review history of their article (what does this mean?). If published, this will include your full peer review and any attached files.

Reviewer #1: Yes: Prof. Edoardo De Robertis - University of Perugia, Italy

Reviewer #2: No

---

## [Author Response · Author response to Decision Letter 0]

6 Jun 2020

We are uploading our response file .

---

## [Decision Letter · Decision Letter 1]

19 Jun 2020

Fluid overload is a determinant for cardiac structural and functional impairments in type 2 diabetes mellitus and chronic kidney disease stage 5 not undergoing dialysis

PONE-D-20-08628R1

Dear Dr. Kim,

We’re pleased to inform you that your manuscript has been judged scientifically suitable for publication and will be formally accepted for publication once it meets all outstanding technical requirements.

Kind regards,

Vincenzo Lionetti, M.D., PhD

Academic Editor

PLOS ONE

Additional Editor Comments (optional):

Reviewers' comments:

Reviewer's Responses to Questions

**Comments to the Author**

1. If the authors have adequately addressed your comments raised in a previous round of review and you feel that this manuscript is now acceptable for publication, you may indicate that here to bypass the “Comments to the Author” section, enter your conflict of interest statement in the “Confidential to Editor” section, and submit your "Accept" recommendation.

Reviewer #1: All comments have been addressed

2. Is the manuscript technically sound, and do the data support the conclusions?

Reviewer #1: Yes

3. Has the statistical analysis been performed appropriately and rigorously? 

Reviewer #1: Yes

4. Have the authors made all data underlying the findings in their manuscript fully available?

Reviewer #1: Yes

5. Is the manuscript presented in an intelligible fashion and written in standard English?

Reviewer #1: Yes

6. Review Comments to the Author

Reviewer #1: Dear Authors, thanks a lot to have evaluated the suggestions/observations raised and accordingly modified the text of the manuscript

7. PLOS authors have the option to publish the peer review history of their article (what does this mean?). If published, this will include your full peer review and any attached files.

Reviewer #1: Yes: Edoardo De Robertis

---

## [Editor Report · Acceptance letter]

13 Jul 2020

PONE-D-20-08628R1 

Fluid overload is a determinant for cardiac structural and functional impairments in type 2 diabetes mellitus and chronic kidney disease stage 5 not undergoing dialysis 

Dear Dr. Kim:

I'm pleased to inform you that your manuscript has been deemed suitable for publication in PLOS ONE. Congratulations! Your manuscript is now with our production department. 

Kind regards, 

on behalf of

Prof. Vincenzo Lionetti 

Academic Editor

PLOS ONE